# Study of the Affinity Law of Energy and Cavitation Characteristics in Emergency Drainage Pumps at Different Rotating Speeds

**Weidong Cao** [1,2,]* and **Jiayu Mao** [1,2]

1   Research Institute of Fluid Engineering Equipment Technology, Jiangsu University, Zhenjiang 212013, China; mjy1006a@163.com
2   China National Research Center of Pumps, Jiangsu University, Zhenjiang 212013, China
*   Correspondence: cwd@ujs.edu.cn; Tel.: +86-1395-281-6468

**Abstract:** The affinity law is widely used in pump design and experiments. The applicability of the affinity law in an emergency drainage pump at different rotating speeds was studied. Experiments and numerical simulation through ANSYS CFX (Computational Fluid Dynamics X) 15.0 software were used to research the affinity law characteristics. Results show that the simulation of characteristics is basically consistent with the experimental curves. In small flow rate conditions, due to the existence of obvious differential pressure between the pressure side and the suction side in the impeller blade tip area, the leakage flow occurs at the tip clearance, which collides with the main stream at the inlet and generates vortices at the leading edge of the impeller. The tip leakage flows of the pump at four different rotating speeds were compared, and it was found that the tip leakage increased with increasing rotation speed, and at the same rotation speed, the tip leakage flow was large in the small flow rate condition, which led to the simulation value of the characteristics being greater than the scaling value. As the flow rate increased, the anti-cavitation performance of the pump became worse and the hydraulic loss was larger, so the pump's performance curve deviated from the scaling curve.

**Keywords:** affinity; emergency drainage pump; rotating speed; cavitation

## 1. Introduction

High-speed emergency drainage pumps have been widely applied in rapid responses to solve urgent water-logging problems because they are of light weight and have powerful water transmission capacity [1]. Similar to ordinary centrifugal pumps, the main design methods of drainage pumps are the affinity law and the velocity coefficient method, but with the in-depth study of the traditional basic theory and design methods, some new design methods have been created, such as the increasing flow rate design method, clogging flow channel design method, non-clogging pump design method, two-phase flow design method, and so on [2]. Some pumps are difficult to experiment on because of their large size and higher rotating speed, the traditional affinity law is still a dominant method that is widely used in the process of designing pump products. This method can shorten the product development cycle; however, the applicability of the affinity law for high-speed, axial-flow emergency drainage pumps is challenged. Under the conditions where the pumps runs at small flow rates and where high lift or cavitation occurs at large flow rates, the similarity of the internal flow in high-speed, axial-flow emergency drainage pumps seems to be quite different from that of ordinary centrifugal pumps, making their internal flow characteristics worth studying.

Shi verified the similarity of the scaling model of a typical submersible well pump with a scaling factor of 0.66, and it was found that the designed pump and the model pump basically showed

consistent variation tendency for their head, efficiency, and shaft power under 0.4–1.6-times rated flow conditions, and the highest efficiency points were both achieved under 1.2-times rated flow rate conditions, which means the predicted performance met similar conversion rules [3]. Si studied the applicability of the similarity law under the condition of gas–liquid two-phase flow in centrifugal pumps, and the comparisons between the experimental and simulated results showed that the greater the inlet gas volume fractions was, the worse the applicability to the similarity law [4]. Naoki studied the air–water two-phase flow performance in a conventional centrifugal pump and found the similarity law of the impeller diameter was experimentally confirmed, even in two-phase flow condition; influences of blade height on air–water two-phase flow performances indicated little difference from the similarity law [5]. Zhu studied the hydraulic characteristics between prototype and model pumps in pumping stations, because the local hydraulic loss was the main part in the inlet and outlet channels of the pumping station, but during the efficiency correction, the local loss had not been separated; in fact, it did not need to be converted and had nothing to do with the scale effect [6]. Chen analyzed the approximation of hydraulic simulation, mechanical constraints, and geometric similarity between the model and the prototype for pumps and pump sets. The hydraulic efficiency expression was deduced in terms of its original definition, where pump and pump-set efficiency conversion formulas were derived, which concerned hydraulic friction loss only or all hydraulic losses, including shape resistance and shock vortex resistance [7].

Zhen studied the rotating stall characteristics of an axial-flow pump. The results showed that there were two reversed-flow field areas in the leading edge close to the shroud and in the trailing edge close to the hub of the blade suction surface under rotating stall conditions, and the dominant frequency under the designed conditions in the impeller inlet and the middle of the guide vane was blade passing frequency. However, the dominant frequency under the designed conditions in the impeller outlet was guide passing frequency, because of rotor–stator interaction [8]. Li studied the flows in an axial-flow pump at the design flow rate for five different tip gap sizes by using large eddy simulation. The results indicated that the head, shaft power, and efficiency of the pump decreased as the tip gap size increased, and the velocity of the tip leakage flow in the tip gap increased gradually along the radial direction; when the tip gap size was larger than 1.0‰ of the impeller diameter, the dominant tip leakage vortex extended to the pressure side of the neighbor blade, and the tip separation vortex and several secondary vortices were also found [9]. Zhang simulated and analyzed the formations of a three-dimensional tip leakage vortex cavitation cloud and the periodic collapse of vortex-induced suction side perpendicular cavitation vortices. The improved turbulence model and the homogeneous cavitation model were validated by comparing the simulation results with an experiment by measuring unsteady cavitation shedding flow around a hydrofoil. The unsteady cavitation cloud occurred near the blade trailing edge and the shapes of the sheet cavitation and normal cavitation fluctuated [10]. Zhang also studied the tip leakage vortex in an axial flow pump under small flow rate conditions. Owing to the large pressure difference between the pressure side and suction side at the tip, the axial velocity of flow in the gap was negative, and the absolute value of the axial velocity increased gradually from the pressure side to the suction side. High-speed photography experimental results showed that as the cavitation number decreased gradually, the leakage vortex cavitation was initiated in the tip clearance under small flow rate conditions [11]. Wang studied the internal flow field in a pump installation, where an unstable pattern behind the pump and significant numbers of vortices occurred in the outlet passage, the pressure distribution was irregular in the middle of the blade suction side, and a high-pressure zone existed on the blade pressure side near the leading edge and on the blade suction side near the trailing edge [12]. Zhou studied the stability of the reverse-power generation of a pumping station; the frequency of the pressure pulsation was influenced by the frequency of the runner and was concentrated at low frequencies. The flow pressure fluctuation in the middle and edge of the inlet of the runner was obvious, and the maximum pressure pulsation occurred in the middle of the runner outlet. The amplitude of the pressure pulsation was nearly 3 times that of the outlet edge of the runner, and it was nearly 2 times the amplitudes of the middle and edge of the inlet of the runner [13].

When the pumps work in off-design conditions, it is easy to observe stall, leakage, cavitation, and other phenomena, which lead to a certain deviation between the scaling value and the simulation value after the affinity law is used. The above scholars have mainly focused on the study of the scaling methods of hydraulic characteristics of the actual pump and the model pump. The affinity law was modified according to improving various losses, such as hydraulic loss and volume loss, and various scaling formulas have been proposed to improve the affinity law of pumps. However, there are few studies on the reasons for the differences between the scaling results and the experimental results after the affinity law is used, particularly the study of the affinity law characteristics in emergency drainage pumps at different rotating speeds. In order to study the influencing factors on the affinity law and to figure out the applicability of the similarity theory at variable operating conditions, commercial ANSYS CFX 15.0 software was adopted to simulate the whole flow field of an emergency drainage pump at different flow rates and rotating speeds. The internal flow characteristics were analyzed, and the simulation results were compared with those of experiments. The distributions of cavitation volume fraction in the impeller under different flow rates and different cavitation numbers, the variations of cavitation flow fields in the blade tip region, and the relationship between leakage loss and affinity law in the pump under different rotating speeds were all analyzed. Through analyzing the leakage flow with small flow rates and cavitation flow with large flow rates, affinity law mechanisms related to energy and cavitation characteristics will be discussed.

## 2. Numerical Model and Method

### 2.1. Flow Control Equations

The basic equations of fluid mechanics include the continuity equation, momentum equations, and energy equation. The fluid flow in pumps changes with time and space, and it is a complex, three-dimensional, and unsteady process. The water transported by the emergency drainage pump can be considered an incompressible medium. The equations adopted are as follows.

The continuity equation is the mathematical expression of the law of mass conservation:

$$\frac{\partial \rho}{\partial t} + \frac{\partial(\rho v_x)}{\partial x} + \frac{\partial(\rho v_y)}{\partial y} + \frac{\partial(\rho v_z)}{\partial z} = 0 \tag{1}$$

where $\rho$ is the fluid density; $t$ is the time; $v_x$, $v_y$ and $v_z$ represent the velocity components in $x$, $y$, and $z$ directions.

The momentum equations, named N-S equations, are the mathematical expression of the law of momentum conservation. Although the N-S equations can describe the motion of turbulence, due to the huge difference between the time and space characteristic scales in turbulent flow fields, it is difficult to solve N-S equations in actual flows. The Reynolds time-averaged equations are generally used, and the tensor expression is as follows,

$$\frac{\partial}{\partial x_j}\left(\rho u_i u_j\right) = -\frac{\partial P}{\partial x_i} + \frac{\partial}{\partial x_j}\left(\mu \frac{\partial u_i}{\partial x_j} - \rho \overline{u'_i u'_j}\right) + S_i \tag{2}$$

where $u_i$ is the Reynolds average velocity; $u'_i$ is the velocity fluctuation; $\mu$ is the kinematic viscosity.

The energy equation, meaning the amount of energy change in the controlling body, is equal to the sum of the heat entering the controlling body minus the heat escaping from the controlling body per unit time, and the equation is expressed as,

$$\frac{\partial(\rho T)}{\partial t} + \frac{\partial(\rho v_x T)}{\partial x} + \frac{\partial(\rho v_y T)}{\partial y} + \frac{\partial(\rho v_z T)}{\partial z} = div\left(\frac{k}{c_P}\text{grad}T\right) + S_T \tag{3}$$

where $T$ is the temperature; $k$ is the heat transfer coefficient of the medium; $c_P$ is the specific heat capacity; $S_T$ is the heat source inside the fluid.

*2.2. Turbulence Mode*

According to the number of differential equations used, the commonly used turbulence models can be divided into the zero-equation model, one-equation model, two-equation model, and multiple-equation model. The two-equation turbulence model is widely used at present. Strong shear stress and a reverse pressure gradient exist in the emergency drainage pump. Therefore, the SST (Shear Stress Transport) $k$-$\omega$ model is selected to close the equations, with its turbulent energy $k$ and turbulent fluctuation frequency $\omega$ equation.

Turbulent kinetic energy $k$ equation:

$$\frac{\partial(\rho k)}{\partial t} + \frac{\partial(\rho k u_i)}{\partial x_i} = \frac{\partial}{\partial x_j}\left[(\mu_m + \frac{\mu_T}{\sigma_k})\frac{\partial k}{\partial x_j}\right] + P_k - \beta'\rho k\omega \tag{4}$$

Turbulent pulsation frequency $\omega$ equation:

$$\frac{\partial(\rho\omega)}{\partial t} + \frac{\partial(\rho\omega u_i)}{\partial x_i} = \frac{\partial}{\partial x_j}\left[(\mu_m + \frac{\mu_T}{\sigma_\omega})\frac{\partial\omega}{\partial x_j}\right] + \alpha\frac{\omega}{k}P_k - \beta\rho\omega^2 \tag{5}$$

The relationship between turbulent viscosity $\mu_T$, turbulent kinetic energy $k$ and turbulent pulsation frequency $\omega$ is,

$$\mu_T = \rho\frac{k}{\omega} \tag{6}$$

where $P_k$ is the turbulence generation rate, and also the pressure generation term produced by the velocity gradient; other variables are obtained using the N-S formula. In addition, the constant terms in the model are $\beta' = 0.09$, $\alpha = 5/9$, $\beta = 0.075$, and $\sigma_k = 1$, $\sigma_\omega = 2$.

*2.3. Hydraulic Model*

The emergency drainage pump studied in this paper is mainly used in urban emergency drainage projects. Its compact structure makes it convenient to be transported and used. As shown in Figure 1, the pump mainly includes the motor, inlet section, impeller, and guide vane. The design parameters are as follows: rated flow $Q_{opt}$ is 220 m³/h, head $H$ is 7.5 m, rated rotating speed is 3800 r/min, power is 15 kW, the specific speed is 756, the impeller blade number is 4, the diameter of the impeller is 131.6 mm, the hub diameter of the impeller is 61.8 mm, the clearance between impeller and drum wall is 0.7 mm, the guide vane blade number is 6, the distance between inlet edge of the guide vane and outlet of the impeller is 5 mm, and the diameter at the outlet of the guide blade is 137 mm.

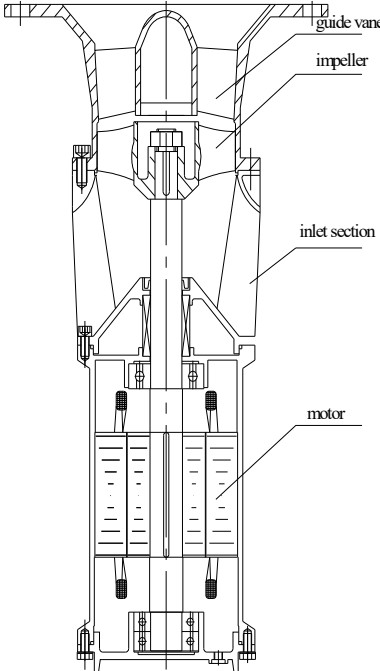

**Figure 1.** Emergency drainage pump.

*2.4. Mesh Division*

Commercial software Icem-cfd(The Integrated Computer Engineering and Manufacturing code for Computational Fluid Dynamics) 15.0 was used to divide the flow field into structured meshes, which are shown in Figure 2. In order to control the mesh distribution in the boundary layer near the wall, the O-type topology is selected. The mesh in the blade boundary layer and its adjacent zone is controlled by the O-grid, and the mesh in the impeller is encrypted. A ten-layer grid is placed in the tip clearance to better capture the flow. As shown in Table 1, in order to obtain the appropriate number of meshes, five grid sizes with different precision of division were compared, and the simulated head and efficiency values were chosen as reference indexes, as shown in Figure 3. When the mesh number reaches over 2.75 million, the simulated results of efficiency ($\eta$) and head ($H$) do not fluctuate obviously with the changing of numbers of meshes, and the total simulation mesh number is 2,756,481.

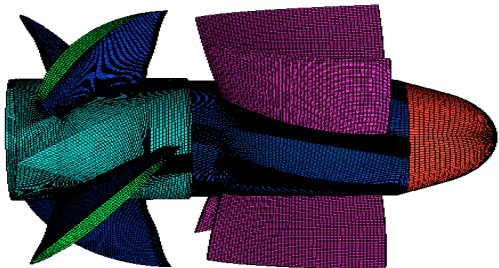

**Figure 2.** Structured mesh in the computational domain.

**Table 1.** Comparison of mesh numbers.

| Scheme | Node Number | Mesh Number |
|:------:|:-----------:|:-----------:|
| 1 | 1983484 | 2021348 |
| 2 | 2302738 | 2527547 |
| 3 | 2624871 | 2756481 |
| 4 | 2995718 | 3122487 |
| 5 | 3327496 | 3463789 |

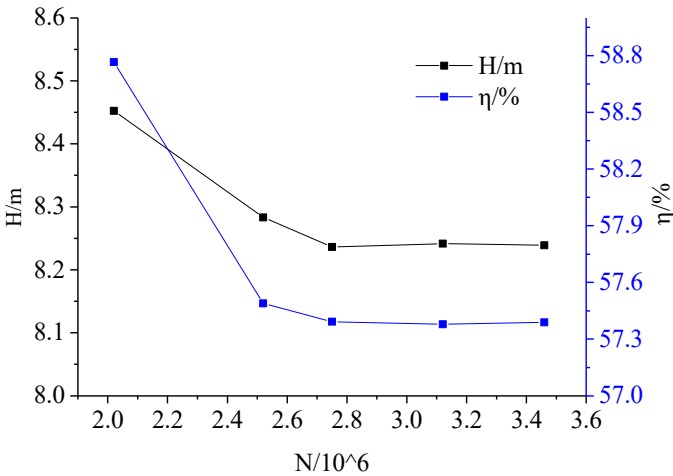

**Figure 3.** Mesh independence analysis.

### 2.5. Simulation Methods and Boundary Conditions

The SST *k*-*ω* turbulent model and CFX 15.0 software were adopted in this paper. A steady numerical simulation was adopted. For the boundary conditions, the inlet was set as the total pressure, while the outlet was set as mass flow rate. The solid wall was set as a non-slip and adiabatic wall surface. The roughness of the main flow passage components, such as the impeller and guide vane, was set at 0.025 mm. The standard wall function was adopted for the near wall surface. The water in the impeller was set as the rotation area, while the rest of the water body was set as the static area, and the outer wall surface of the impeller was set as the counter-rotating wall. The coupling interfaces between the rotating areas and the static areas were set in transient frozen rotor mode, and the coupling interfaces between the static regions were not set in any mode. The convergence accuracy was set at $10^{-5}$ with the higher-order solution format.

The default ZGB(Zwart-Gerber-Belamri) cavitation model was chosen to simulate the cavitation flow, total pressure was selected as the inlet boundary condition, the volume fraction of the liquid phase was set to 1, the volume fraction of the gas phase was set to 0, and mass flow rate was selected as the outer boundary condition. The medium for computational fluid was 25 °C water. The critical cavitation pressure was set to 3574 pa according to the temperature of the saturated steam pressure, the average diameter of cavity was set to $10^{-6}$ mm, and the convergence residual was set to $10^{-5}$.

## 3. Characteristics and Analysis

### 3.1. Characteristics Comparison

The experimental scheme and the tested pump are shown in Figure 4. The main instruments included the LWGY-150A0A3T turbine flow meter, the precision level for which was 0.5%; the $\phi$150 flow rate control valve; and the WT2000 pressure transmitter whose precision level was 0.2%. The precision level for the current and voltage sensors was 0.5%, and the HSJ-2010 hydraulic mechanical comprehensive test instrument was used, the precision level for which was 0.2%.

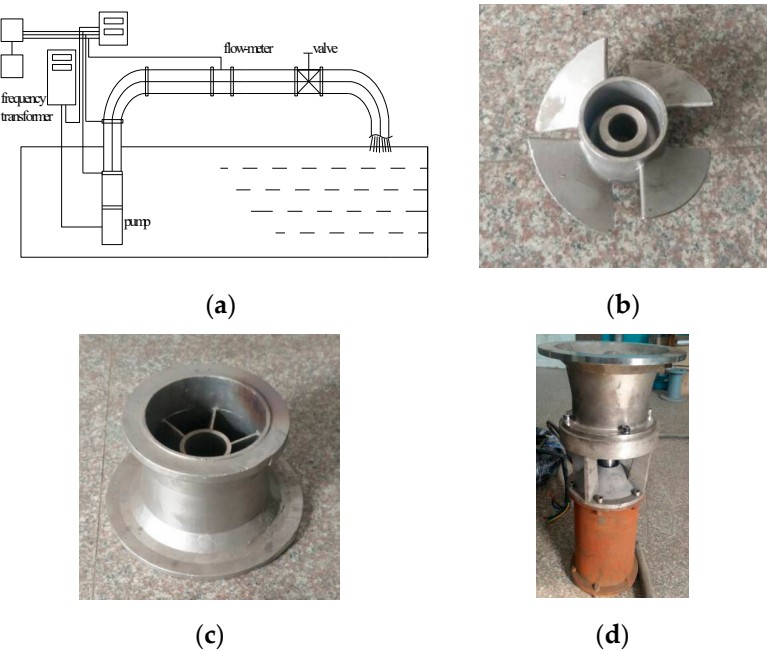

**Figure 4.** Experimental scheme and the tested pump: (**a**) experimental scheme; (**b**) impeller; (**c**) guide vane; (**d**) pump.

As shown in Figure 5, the simulation results are basically consistent with the variation trend of the experiment curves according to the performance comparison of the emergency drainage pump at the design rotating speed of 3800 r/min. The simulation results are almost same as those of the experiments at low flow rates. When the flow rate changes to larger than 300 m³/h, the head and efficiency of the simulation results are gradually higher than those of the experiments. At the design flow rate ($1.0Q_{opt}$), the relative head and efficiency errors between the simulation and experiment are 2.2% and 1.5%, respectively. The head increases when the flow rate decreases from $1.0Q_{opt}$ to $0.8Q_{opt}$, but the change is small. As the flow rate continues to decrease from $0.8Q_{opt}$ to $0.6Q_{opt}$, the head shows a significant increasing trend. However, at large flow rates of $1.0Q_{opt}$ and $1.2Q_{opt}$, the experimental results and the simulation results deviate greatly, and the simulation values of head and efficiency are greater than the experimental values.

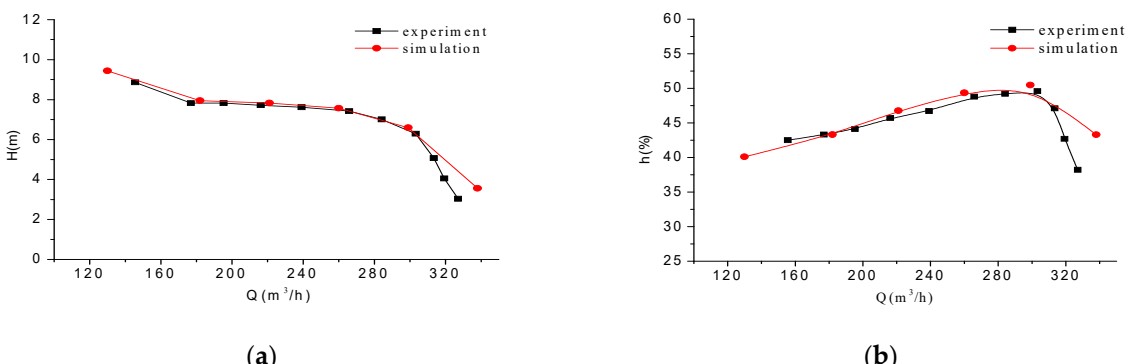

**Figure 5.** Comparison of performance curves: (**a**) *Q-H*(Flow rate-Head) curve; (**b**) *Q-η* (Flow rate-Efficency) curve.

### 3.2. Analysis of the Applicability of the Similarity Theory

The similarity theory is widely used in the pump design process and in experiments. Because the similarity theory is based on a hypothesis, it is empirical; the similarity theory can only be used conditionally in practical applications. The results show that if the size and speed of the model

pump and real pump are similar, when two similar geometric pumps work in similar conditions, their efficiency can be thought of as equal. If the corresponding size of the pumps is equal, or for the same pump and the same transmission medium, the affinity law can be simplified as,

$$Q/Q' = n/n', \ H/H' = (n/n')^2, P/P' = (n/n')^3 \qquad (7)$$

where $Q$ is the flow rate in m$^3$/h, $n$ is the pump speed in r/min, $H$ is the pump head in m, and $P$ is the pump shaft power in kW. The above formulas, known as scaling laws, represent the relationship between the performance parameters when the pump speed changes. During the experiments, the pump speed changes under different working conditions; in general, the data at each experimental speed must be converted to the rated speed. The scaling method relies on the above scaling laws.

The high-speed emergency drainage pump studied in this paper will be applied in different conditions by adjusting the rotating speeds. Based on the experimental data and a rotating speed of 3800 r/min, similar flow rates and heads at rotating speeds of 3600 r/min, 3400 r/min, and 3200 r/min are obtained through Equations (7). Additionally, the true experiment results at rotating speeds of 3600 r/min, 3400 r/min, and 3200 r/min are given and compared, as shown in Figure 6a. The experiment results at rotating speeds of 3600 r/min and 3400 r/min are consistent with the corresponding scaling results under design flow rates, and the experimental value is slightly larger than the scaling value for small flow rates. However, when the rotating speed drops to 3200 r/min, there is a large deviation between the experimental result and the scaling result, the experimental values are far greater than the scaling values, indicating that the scaling of the affinity law has a certain scope of application. Beyond this scope, the affinity law will not be applicable.

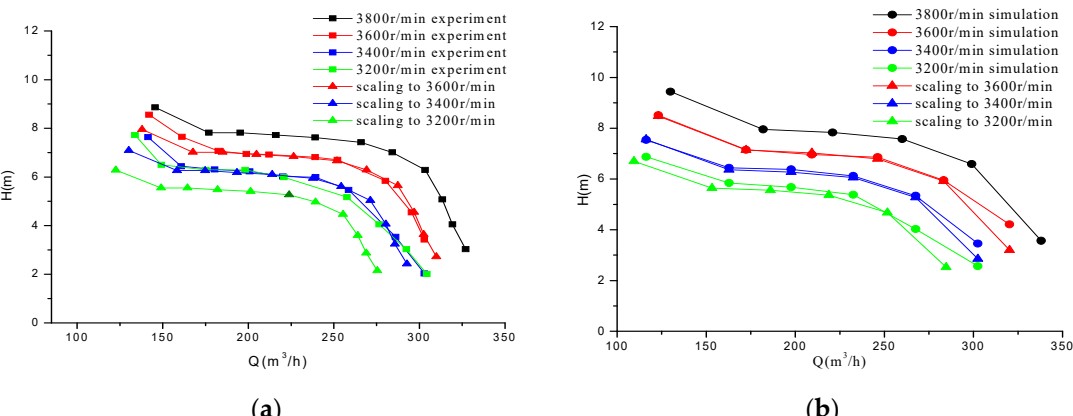

**Figure 6.** Comparison of head curves: (**a**) experimental scaling results and comparison; (**b**) simulation scaling results and comparison.

The simulation heads at rotating speeds of 3800 r/min, 3600 r/min, 3400 r/min, and 3200 r/min are shown in Figure 6b. Based on the simulation head at the rotating speed of 3800 r/min, the heads at rotating speeds of 3600 r/min, 3400 r/min, and 3200 r/min are also obtained through Equation (7). The trends in Figure 6b are generally consistent with those in Figure 6a. When the rotating speed drops to 3200 r/min, the simulation result and the scaling result also show great deviation. This indicates that the affinity law can only be applied to the emergency drainage pump within the range of a 15% speed reduction.

*3.3. Internal Flow Field Analysis*

3.3.1. Surface Pressure Distribution of the Impeller

Figure 7 shows the static pressure on the impeller blades one the pressure side at different flow rates and a rotating speed of 3800 r/min. In general, the pressure on the pressure side of the

blades increases gradually from the inlet to the outlet and from the hub to the rim. However, the pressure distribution under the flow rate of $0.8Q_{opt}$ does not apply to this rule. As shown in Figure 7a, the pressure on the pressure side gradually decreases from the inlet side to the outlet side, and a relatively high-pressure area forms in the inlet area. As the flow rate increases, the maximum pressure and the high-pressure area gradually decrease, and the pressure gradient on the blades becomes more obvious. At the flow rate of $1.2Q_{opt}$, the low pressure area begins to appear in the inlet area on the pressure side, and the low pressure area gradually expands with the further increase of the flow rate.

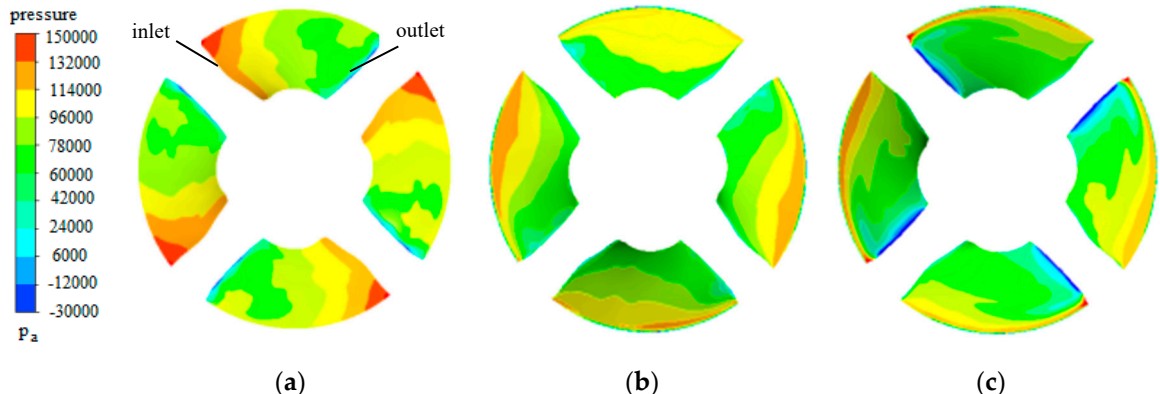

|           |           |           |
|:---------:|:---------:|:---------:|
| (**a**)   | (**b**)   | (**c**)   |

**Figure 7.** Static pressure on impeller blades on the pressure side: (**a**) $0.8Q_{opt}$; (**b**) $1.0Q_{opt}$; (**c**) $1.2Q_{opt}$.

Figure 8 shows the static pressure on the impeller blades on the suction side at different flow rates and a rotating speed of 3800 r/min. The static pressure on the suction side increases gradually from the inlet to the outlet. At the flow rate of $0.8Q_{opt}$, an obvious low pressure zone forms in the area near the inlet edge and on the rim near the inlet. This verifies that the cavitation phenomenon in the impeller of the emergency drainage pump usually happens at the inlet suction side and in tip clearance area. With the increase of the flow rate, the static pressure of the inlet area gradually increases. When the flow rate is $1.2Q_{opt}$, almost the entire outer edge of the blade becomes a relatively high-pressure area.

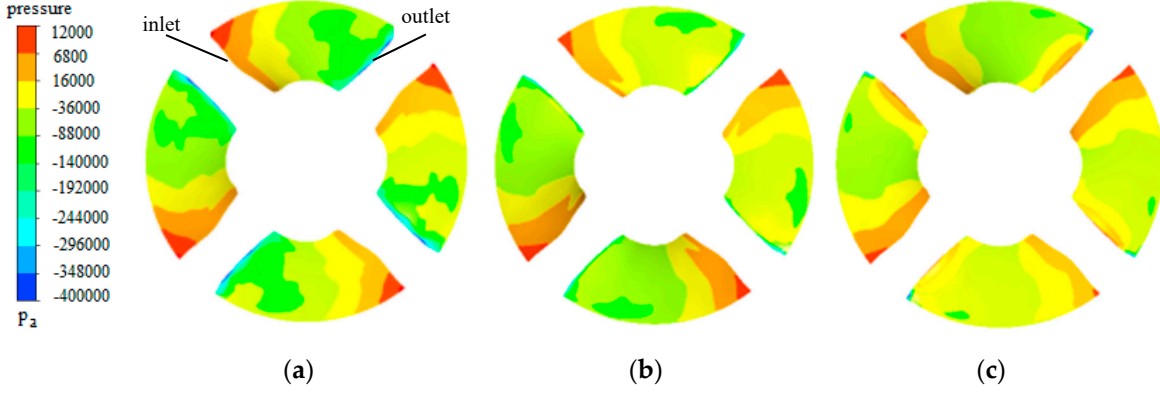

|           |           |           |
|:---------:|:---------:|:---------:|
| (**a**)   | (**b**)   | (**c**)   |

**Figure 8.** Static pressure on impeller blades on the suction side: (**a**) $0.8Q_{opt}$; (**b**) $1.0Q_{opt}$; (**c**) $1.2Q_{opt}$.

Comparing the pressure distribution, it can be seen that the static pressure on the pressure side and the suction side near blade inlet will change with the alternations of flow rate. When the pump works at the design flow rate, the flow path conforms to the blade shape, and the shock loss can be neglected. When the pump performs under low flow rate, a negative angle exists between the flows and the blade, flow separation on the suction side happens, and a low pressure area appears at the inlet area on the suction side. When the pump performs under a higher flow rate, a positive angle exits between the flows and the blade, flow separation occurs on the blade pressure side, and a low pressure area appears at the inlet area on the pressure side.

### 3.3.2. Streamline between Impeller Blades

Because there is a clearance between the impeller and the cylinder wall, due to the pressure difference between the blades on the pressure and suction side, leakage flow occurs at the tip region and interacts with the incoming flow, then vortices form, which result in a very complex flow in the tip regions.

As shown in Figure 9, the radial coefficient *span* of the blade is defined as the dimensionless distance of the blade from the hub along the radial direction to the edge of the wheel rim.

$$span = (r - r_h)/(r_t - r_h) \tag{8}$$

where $r$ is the radius in mm, $r_t$ the impeller rim radius in mm, and $r_h$ the impeller hub radius in mm.

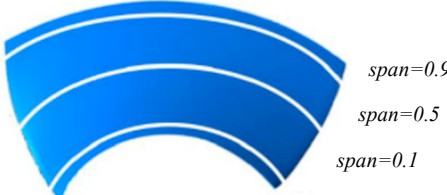

**Figure 9.** Impeller blades with different radial coefficients.

Figure 10 shows the streamline on the expansion plane as *span* = 0.5 in the impeller and guide vane, the rotating speed is 3800 r/min, and the flow rate is $0.8Q_{opt}$. Flow separation happens at the inlet edge of the impeller blade on the suction side and forms a low pressure area. There are strong vortices near the guide vane blades on the suction side. With the development of these new vortices, the vortices gradually fall off from the guide vane blades and spread to whole outlet. At the flow rate of $1.0Q_{opt}$, there is no flow separation in the impeller and the flow is stable. There are a few vortices between the guide vane blades on the suction side near the outlet. At the flow rate of $1.2Q_{opt}$, the flow separation is relatively weak and occurs on the guide vane blades on the pressure side in the inlet area.

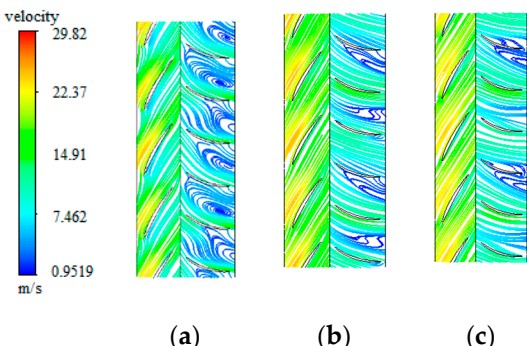

**Figure 10.** Streamline in the pump: (**a**) $0.8Q_{opt}$; (**b**) $1.0Q_{opt}$; (**c**) $1.2Q_{opt}$.

### 3.3.3. Turbulent Kinetic Energy

The turbulent kinetic energy represents the mechanical energy loss of the fluid. Figure 11 shows the distribution of turbulent kinetic energy at the axial section of the pump at the flow rates of $0.8Q_{opt}$, $1.0Q_{opt}$, and $1.2Q_{opt}$ respectively, at a rotating speed of 3800 r/min. The turbulent kinetic energy in the pump is mainly distributed in the impeller, and it gradually decreases as the flow rate increases from $0.8Q_{opt}$ to $1.2Q_{opt}$. Therefore, the flow in the impeller is the most disordered, and the hydraulic loss is the largest with small flow rates.

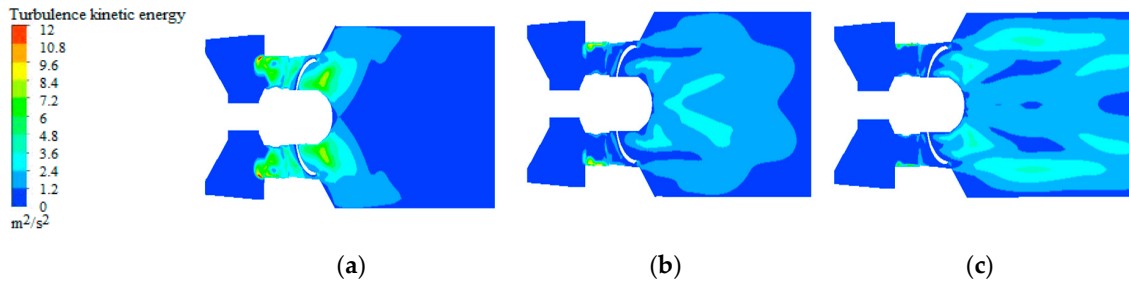

**Figure 11.** Distribution of turbulent kinetic energy at different flow rates: (**a**) $0.8Q_{opt}$; (**b**) $1.0Q_{opt}$; (**c**) $01.2Q_{opt}$.

Figure 12 shows the distribution of turbulent kinetic energy in the impeller at different rotating speeds, with a flow rate of $1.0Q_{opt}$. It can be seen from the figure that the turbulent kinetic energy gradually increases with increasing rotating speed at the same flow rate. The maximum turbulent kinetic energy appears in the rim region, because the pressure difference between the impeller pressure and the suction surfaces causes leakage and secondary flow at the clearance, and the interaction with the incoming flow forms vortices, which result in large hydraulic loss.

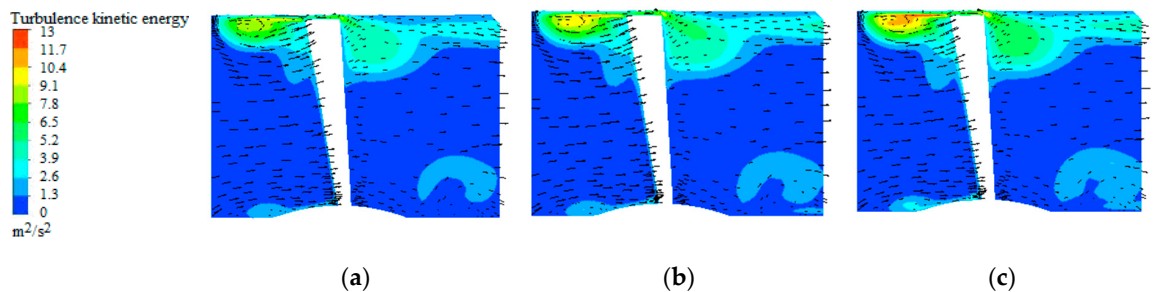

**Figure 12.** Turbulent kinetic energy distributions at different rotating speed, at a flow rate of $1.0Q_{opt}$: (**a**) 3400 r/min; (**b**) 3600 r/min; (**c**) 3800 r/min.

### 3.3.4. Clearance Leakage

Leakage flow exists in the clearance between the impeller rim and the cylinder wall. This is the volume of fluid passing through the tip gap section in unit time, which can be obtained by integrating the velocity component of the normal direction with the square value of the section. The leakage amount in the unit area of the feature section is defined as follows:

$$Q_1 = \int_A v_i ds \tag{9}$$

where $v_i$ is the normal velocity through the characteristic section in m/s; $A$ is the area of the characteristic section in m$^2$. The characteristic section is defined as the plane formed by two axial sections of the impeller and the extension surfaces of the pressure side and suction side, the impeller rim face, and the wheel chamber wall surface. The area containing the cross shaft and blade center is taken as the reference section. Based on this reference section, eight other characteristic sections are recorded by turning left or right by 10°, 20°, 30°, and 35°. A total nine characteristic sections are named as I, II, . . . IX, as shown in Figure 13.

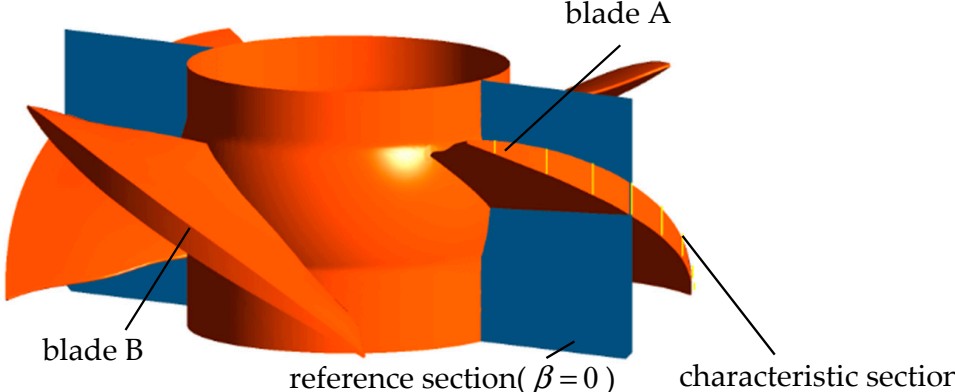

**Figure 13.** Characteristic sections.

Figure 14 shows the variation trends of tip clearance leakage at different flow rates, with a rotating speed of 3800 r/min. Since the area of the characteristic section increases first and then decreases with the change of the chord length coefficient, the leakage increases first and then decreases, forming a parabolic-shaped line. Because of the large pressure difference, the maximum leakage occurs near the reference section. The total amount of leakage under the flow rate of $0.7Q_{opt}$ is the largest. The flows in the tip clearance area are more complex and it is easy for back-flow to occur, leading to large hydraulic loss. As the flow rate increases, the total amount of tip leakage decreases, the tip leakage is more uniform with the change of chord length coefficient, and the energy loss is relatively small.

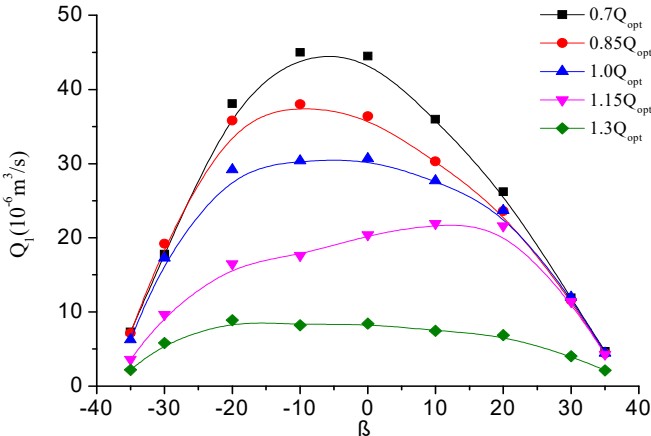

**Figure 14.** Leakage at different flow rates.

Figure 15 shows the variation trends of tip clearance leakage at different rotating speeds, with a flow rate of $1.0Q_{opt}$. The leakage increases first and then decreases with the change of chord length coefficient at various rotational speeds, and it gradually increases with the increase of rotating speed. The leakage reaches the maximum at the rotating speed of 3800 r/min. The blade tip leakage at different rotating speeds is disproportionate. As a result, the simulation value of the head at each speed is slightly higher than the scaling value; that is to say, the leakage does not meet the affinity law.

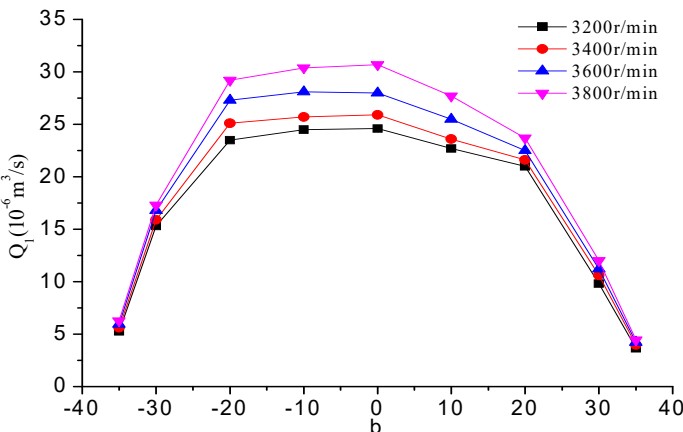

**Figure 15.** Leakage at different rotating speeds.

## 4. Analysis of Cavitation

### 4.1. Cavitation Characteristic Curve

The ZGB model is used for the cavitation model, and its expression is as follows,

$$m = \begin{cases} F_v \dfrac{3\alpha_{ruc}(1-\alpha_v)\rho_v}{R_B} \sqrt{\dfrac{2}{3}\dfrac{|p_v-p|}{\rho_l}} & (\ if\ p < p_v) \\[12pt] F_c \dfrac{3\alpha_v\rho_v}{R_B} \sqrt{\dfrac{2}{3}\dfrac{|p-p_v|}{\rho_l}} & (\ if\ p > p_v) \end{cases} \tag{10}$$

where $p_v$ is the vaporization pressure of the liquid; $R_B$ is the radius of a single bubble; $a_v$ is the volume fraction of the gas; $\rho_l$, $\rho_v$ are the density values of fluid and steam, respectively; the empirical condensation coefficient is $F_c = 0.01$, and the evaporation coefficient is $F_v = 50$.

The cavitation characteristic of the emergency drainage pump was numerically simulated through altering the inlet pressure. The cavitation characteristic curve was obtained at the flow rate of $1.0Q_{opt}$ and rotating speed of 3800 r/min, as shown in Figure 16. NPSH$_a$ is defined as,

$$NPSH_a = \frac{p_{in} - p_v}{\rho g} + \frac{v_{in}^2}{2g} \tag{11}$$

where $p_{in}$ is the static pressure at the pump inlet in p$_a$; $p_v$ is the saturated vapor pressure, 3574 p$_a$; $\rho$ is the density in kg/m$^3$; $g$ is the acceleration of gravity in m/s$^2$; $v_{in}$ is the velocity at the pump inlet in m/s. When the NPSH$_a$ is higher than 12 m, the head remains almost the same and no cavitation occurs. With the decreasing value of NPSH$_a$, the head begins to change. If $10 <$ NPSH$_a < 12$, due to a slight development of cavitation on the blades, the head slightly increases. The cavitation in the impeller starts but only a few bubbles exist, and the influence on the flow is small. If NPSH$_a < 10$, the head drops obviously. When NPSH$_a = 9.75$ m, the drop in head reaches about 3%. When NPSH$_a$ decreases further, the head drops sharply. In general, the critical NPSH$_a$ is defined as the head dropping by 3%. The emergency drainage pump's critical NPSH$_a$ is value 9.75 m. Cavitation will not occur under submersible conditions at the design flow rate and rated rotating speed, however, this does not mean the pump is of good cavitation character. The compact structure design method and high rotating speed bring about a common cavitation character.

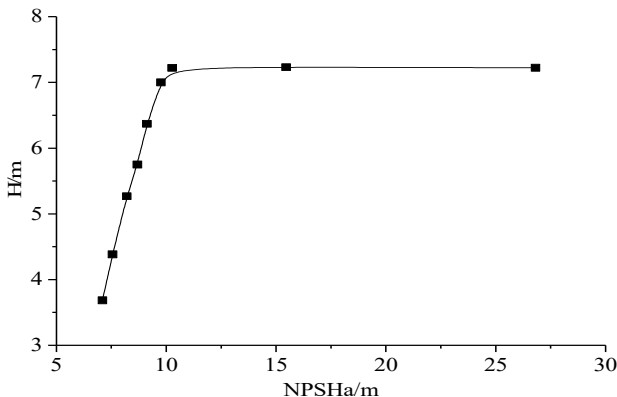

**Figure 16.** Cavitation performance at 3800 r/min and a flow rate of $1.0Q_{opt}$.

### 4.2. Cavitation Bubbles in the Impeller

Figure 17 shows the distribution of cavitation volume fraction on the surface of the impeller at a design flow rate of $1.0Q_{opt}$, at a rotating speed of 3800 r/min. The white part represents the contour with a cavitation vapor volume fraction of 0.1. If $u_2$ is the impeller rim rotating velocity in m/s, the cavitation number $\sigma$ is defined as,

$$\sigma = \frac{p_{in} - p_v}{\rho u_2^2 / 2} \tag{12}$$

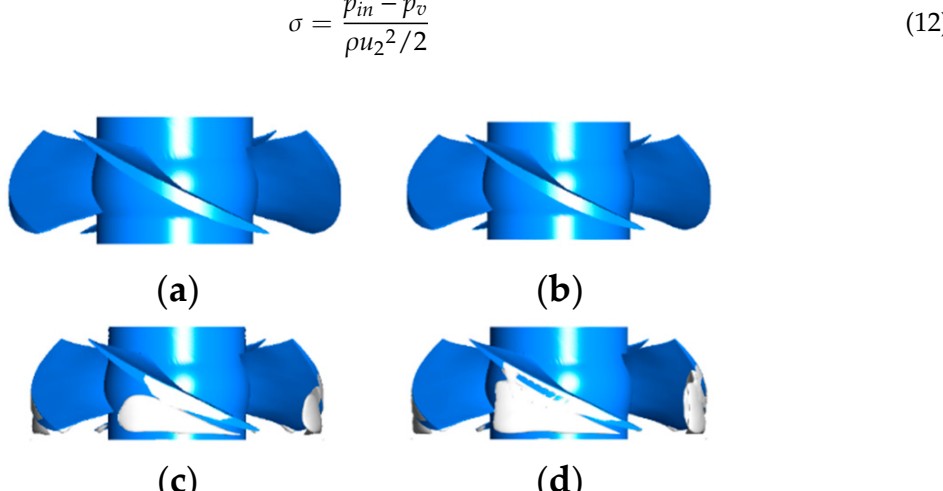

**Figure 17.** Vapor volume distribution on the impeller with different cavitation numbers: (**a**) σ = 1.16; (**b**) σ = 0.58; (**c**) σ = 0.43; (**d**) σ = 0.37.

When σ equals 1.16 or 0.58, the inlet pressure is relatively high and there is almost no distribution of cavitation bubbles on the surface of the impeller. When the inlet pressure decreases to σ = 0.43, a small amount of bubbles begin to appear near the rim of the impeller blades on the suction side close to the inlet. As the pressure further decreases, bubbles diffuse towards the outlet edges of the blades and the hub side. As σ = 0.37, the middle areas between blades are of low pressure, but there are no obvious cavitation bubbles. These phenomena indicate that cavitation of this emergency drainage pump mainly occurs in the tip clearance area of the impeller under the design flow conditions, manifesting as angular vortex cavitation, entrainment area cavitation, and tip leakage vortex cavitation. Among them, angular vortex cavitation and tip leakage vortex cavitation occur firstly in the tip clearance area. With the reduction of the cavitation number, the tip leakage vortex cavitation appears in the entrainment area near the suction side, and the cavitation area and cavitation bubbles increase rapidly.

Three cylindrical cross-sections of *span* are built at values of 0.75, 0.8, and 0.95 near the impeller blade rim. The cavitation vapor volume fractions are shown in Figure 18. As σ = 0.4, no cavitation vapor appears when the cross-section of *span* = 0.75, however, a small amount of cavitation vapor appears when the cross-section of *span* = 0.8, and a large amount of cavitation vapor appears when the

cross-section of *span* = 0.95. This indicates that the cavitation forming in the impeller is tip leakage vortex cavitation.

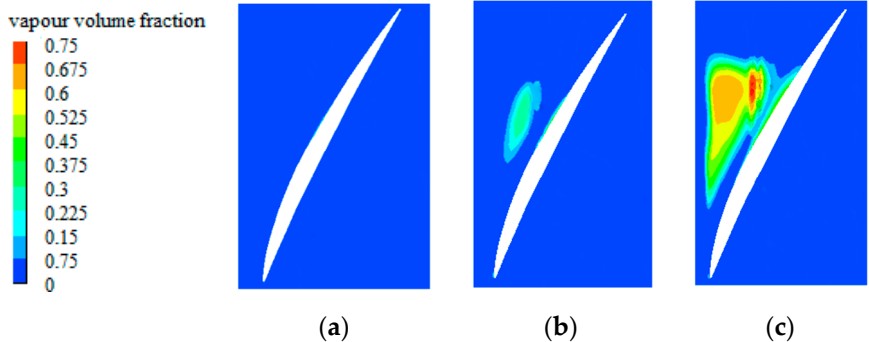

**Figure 18.** Vapor volume distribution on the impeller with different cross-section values: (**a**) *span* = 0.7; (**b**) *span* = 0.8; (**c**) *span* = 0.95.

### 4.3. Cavitation Performance at Different Flow Rates

Figure 19 shows the simulation cavitation characteristic curves of the emergency drainage pump at three different flow rates with a rotating speed of 3800 r/min. The critical $NPSH_a$ values are 6.42 m, 9.75 m, and 14.76 m, respectively, in the order of flow rate from low to high. The minimum critical $NPSH_a$ value for the three flow rates occurs at $0.7Q_{opt}$, which indicates that anti-cavitation performance is relative better with low flow rates. As the flow rate increases, the corresponding critical $NPSH_a$ value also increases, which means that the anti-cavitation performance of the pump decreases. It can be seen that the cavitation performances of the pump cannot meet the affinity law at different flow rates.

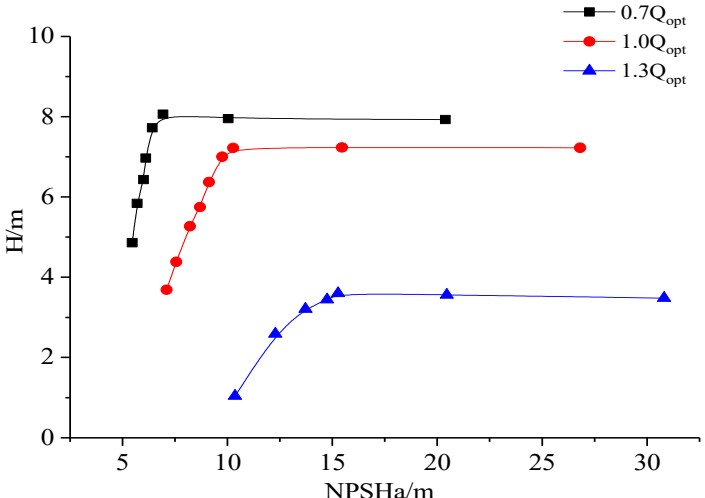

**Figure 19.** Cavitation performance at different flow rates.

Figure 20 shows the cavitation volume fraction of 0.1 in the impeller at the critical $NPSH_a$ value. The flow rates are $0.8Q_{opt}$, $1.0Q_{opt}$, and $1.2Q_{opt}$, and the rotating speed is 3800 r/min. Under a small flow rate of $0.8Q_{opt}$, not only there is leakage vortex cavitation and entrainment zone cavitation in the pump, but there is also attached cavitation on the suction side. At the design flow rate of $1.0Q_{opt}$, the cavitation vortex rope has a slender shape compared with those at small flow rate, and the vane tip leakage vortex cavitation and entrainment zone increase. At a large flow rate of $1.2Q_{opt}$, the cavitation gradually favors the suction side, and the attached cavitation occurs on the suction surfaces of the leading edge, which connects with leakage vortex cavitation and entrainment area cavitation. This is because the main flow forms the separation area on the blades' suction surfaces, leading to the weak cavitation resistance of the pump, especially with large flow rates. The head decreases with the

deterioration of cavitation performance. Cavitation is prone to occur at high flow rates, which results in small deviations between the simulation value and the scaling value. As a result, the cavitation performances of the pump cannot meet the affinity law under different flow rates.

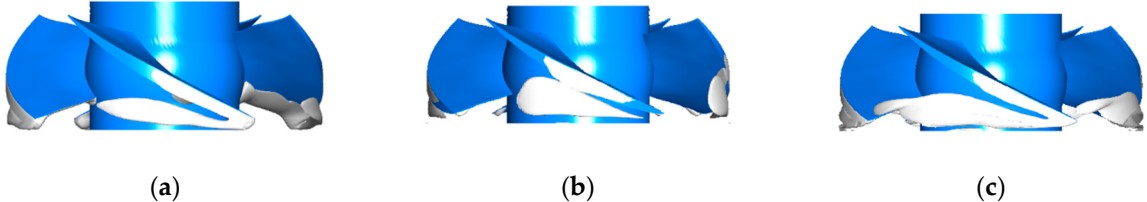

(**a**)             (**b**)             (**c**)

**Figure 20.** Cavitation distributions on the impeller at different flow rates: (**a**) $0.8Q_{opt}$; (**b**) $1.0Q_{opt}$; (**c**) $1.2Q_{opt}$

The Q-H characteristics of $1.15Q_{opt}$ and $1.3Q_{opt}$ at rotating speeds of 3600 r/min and 3400 r/min were simulated by adding the cavitation model. Based on the simulation result at 3800 r/min, similar flow rates and heads at rotating speeds of 3600 r/min and 3400 r/min are obtained again through Equation (7). Scaling and simulation results are all shown in Figure 21. Under large flow rates, the head simulation results at rotating speeds of 3600 r/min and 3400 r/min decrease significantly compared with Figure 6b, and the simulation results are closer to the scaling results. However, there are still some deviations. It is proven that the affinity law is slightly affected by cavitation at large flow rates.

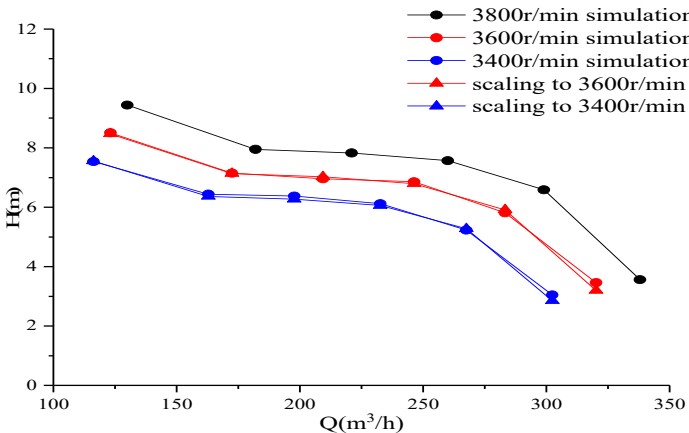

**Figure 21.** Q-H curves of the simulation with the cavitation model.

## 5. Conclusions

Through the comparison of experiments and numerical simulations, the flow details for the emergency drainage pump operating at high speed under the conditions of small and large flow rates are shown, along with cavitation. The reason for the deviation in similar theories for this type of pump is analyzed.

(1) For small flow rates, the static pressure on the pressure side of the blade is significantly higher than that on the suction side, and the static pressure from the inlet to the outlet of the impeller blade shows an upward trend. Because of the obvious pressure difference between the pressure side and the suction side in the tip area, there is tip leakage flow in the tip clearance area, which collides with the main flow at the inlet, generating an eddy current at the leading edge of the impeller and generating greater turbulent kinetic energy in the tip area. Leakage has a great influence on the external characteristic curve, especially at small flow rates.

(2) For large flow rates, with the decrease of $NPSH_a$, the main impeller cavitation is the tip leakage cavitation. The hydraulic performance of the emergency drainage pump is affected by cavitation.

(3)　　If the tip leakage is considered and the cavitation model is used for simulation, the affinity law can be improved a lot. The affinity law can only be applied to the emergency drainage pump within the range of up to 15% speed reduction

**Author Contributions:** W.C. designed and tested the scheme, performed simulations, and organized the paper. J.M. helped to analyze the data.

**Funding:** This work was supported by the National Key R&D Program of China (2018YFC0810506) and the Key R&D Program of Zhenjiang (SH2017049).

**Conflicts of Interest:** The author(s) declared no potential conflicts of interest with respect to the research, authorship, and/or publication of this article.

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
