# Peer review of "Study of the Affinity Law of Energy and Cavitation Characteristics in Emergency Drainage Pumps at Different Rotating Speeds"

_processes, doi:10.3390/pr7120932_

Round 1
Reviewer 1 Report
The manuscript describes a CFD study on an emergency drainage pump at different rotating speeds, with a specific focus on the applicability of the affinity law.
The following are the major remarks:
- English language is poor. The text is full of grammatical mistakes and typos (bad spaces, bad capitals, etc). This aspect requires a deep and careful revision
- Introduction needs to be improved. In particular, it is necessary to underline the novelty and the originality of the proposed study and how the work may be useful to the research community
- Eq. (3) needs to be revised
- In general, quality of the figures is poor. Different fonts are adopted within the same figure. Please, revise it.
- In Section 3 some experimental results are adopted in order to give strength to the numerical results. However, no info regarding test bench and measurement system are provided. Please, add a section/subsection regarding experimental tests
Reviewer 2 Report
After my review I have seen that the paper is well written, and it is very illustrated from the learning point of view.
From the point of view of the investigation I have not detected any improvement with respect to what is already known, in the introduction they say some things but references are not spaced and not discussed.
It should be made clear in the abtract and in the conclusions.
From my point of view, Ansys figures are widely used and some geometric figures are very obvious and not necessary.
I consider it necessary to make clear which is the acatual state of knowledge and which is the knowledge gap that is covered.
